# Short-Chain Fatty Acids and Their Metabolic Interactions in Heart Failure

**DOI:** 10.3390/biomedicines13020343

**Published:** 2025-02-03

**Authors:** Laura Chulenbayeva, Argul Issilbayeva, Aliya Sailybayeva, Makhabbat Bekbossynova, Samat Kozhakhmetov, Almagul Kushugulova

**Affiliations:** 1National Laboratory Astana, Nazarbayev University, Astana 010000, Kazakhstan; argul.issilbayeva@nu.edu.kz (A.I.); skozhakhmetov@nu.edu.kz (S.K.); akushugulova@nu.edu.kz (A.K.); 2Heart Center, CF “University Medical Center”, Astana 010000, Kazakhstan; s.aliya@umc.org.kz (A.S.); m.bekbosynova@umc.org.kz (M.B.)

**Keywords:** heart failure, gut microbiome, short-chain fatty acids, NT-proBNP, left ventricular ejection fraction

## Abstract

Short-chain fatty acids (SCFAs), produced through fermentation of dietary fibers by gut bacteria, play a central role in modulating cardiovascular function and heart failure (HF) development. The progression of HF is influenced by intestinal barrier dysfunction and microbial translocation, where SCFAs serve as key mediators in the gut–heart axis. This review examines the complex metabolic interactions between SCFAs and other gut microbiota metabolites in HF, including their relationships with trimethylamine N-oxide (TMAO), aromatic amino acids (AAAs), B vitamins, and bile acids (BAs). We analyze the associations between SCFA production and clinical parameters of HF, such as left ventricular ejection fraction (LVEF), N-terminal pro-B-type natriuretic peptide (NT-proBNP), and glomerular filtration rate (GFR). Gaining insights into metabolic networks offers new potential therapeutic targets and prognostic markers for managing heart failure, although their clinical significance needs further exploration.

## 1. Introduction

Chronic heart failure (CHF) represents a global healthcare issue associated with high medical expenditures, high morbidity, and mortality. Moreover, its prevalence is growing and has become a problem for millions of people worldwide, with over 26 million individuals suffering from this condition [1]. The incidence of HF varies and increases with age, especially among the elderly [2]. HF has impacted 2.01% of individuals, with an average age of 75.2 years, 48.8% having coronary artery disease, and 34.5% having diabetes. Among 51,442 patients with ejection fraction (EF) assessments, 39.1% observed a decline. Of over 100,000 patients with an estimated glomerular filtration rate (eGFR) recorded, 49% had stage III–V chronic kidney disease (CKD) [3]. Major risk factors include hypertension, ischemic heart disease, diabetes, obesity, smoking, and prior myocardial infarction. This condition is associated with high mortality, and despite treatment, the prognosis remains unfavorable, with recurrent hospitalizations and deterioration in the quality of life for many patients [4,5]. It represents symptoms and/or signs caused by the heart’s structural and/or functional abnormalities, accompanied by impairment of compensatory mechanisms and pathogenic processes [6]. HF can manifest as an acute form, associated with circulating markers involved in the development and progression of HF, such as chronic disease characterized by altered inflammatory status driven by proinflammatory mediators, which play a crucial role in its development [7].

The importance of gut microbiota metabolites, particularly SCFAs, in HF is driven by their role in modulating various physiological processes that impact cardiac function and overall cardiovascular health. SCFAs, including acetate, propionate, and butyrate, are produced through the fermentation of dietary fibers by gut bacteria [8,9]. Understanding the relationship between gut microbiota metabolites, particularly SCFAs, and HF is crucial for comprehending the complex interplay between gut health and cardiovascular function. SCFAs, derived from the fermentation of dietary fibers by gut bacteria, have become key mediators in these relationships [10,11].

A study based on an experimental mouse model of myocarditis evaluated the composition of the gut microbiota at different stages and varying degrees of severity of heart dysfunction. Additionally, comprehensive investigations were conducted into the associations between the gut microbiome and circulating metabolites in the body [12,13]. In the scientific literature, the LVEF is often recognized as a crucial parameter for assessing cardiac function in HF. Its association with SCFAs may be insufficiently documented, as SCFAs are more commonly linked to gut microbiota and metabolic health rather than directly to cardiac function. However, emerging research indicates that the gut microbiota and its metabolites, including SCFAs, may indirectly impact cardiovascular health through systemic inflammation, metabolism, and other pathways. Further investigation into the potential association between LVEF and SCFAs would be valuable for a comprehensive understanding of their interaction with CVD [13].

Studies have indicated that the gut microbiota and its byproducts could impact the onset of HF and related risks. Metabolites associated with the gut microbiota might affect multiple bodily systems, including the cardiovascular system. Notably, SCFAs have positive effects on health, and HF patients show a decrease in bacteria involved in SCFA production. The aim of this review is to explore the relationship between gut microbiome metabolites and clinical parameters of HF, with a particular focus on SCFAs. Essentially, the convergence of published data underscores the critical role of metabolites derived from the gut microbiome in the pathophysiology of HF and highlights the potential for targeted therapeutic interventions and promise as risk prognostic factors to alter the management of this complex condition. However, their clinical significance has not yet been fully assessed and requires further study.

## 2. Gut Microbial Metabolites in Heart Failure

### 2.1. Production and Role of SCFAs

SCFAs are formed through the fermentation of dietary fibers and resistant starches by gut bacteria. As SCFAs are the predominant anions in the lumen of the large intestine, they are absorbed and partly metabolized by intestinal epithelial cells and transported into the bloodstream. Once in circulation, they interact with host proteins and influence physiological processes [14,15]. Bacterial fermentation results in the creation of SCFAs like acetate, propionate, and butyrate. These SCFAs positively impact colon health or improve gut barrier integrity, promoting intestinal tissue growth [16]. Also, SCFAs can improve glucose and lipid metabolism, reduce blood cholesterol levels [17,18,19], and modulate immunity and inflammatory responses [20,21], aiding in disease prevention [17,22,23].

According to Carley et al., SCFAs play a pivotal role in HF due to more effective oxidation, precisely on the mitochondrial level, yet surpassing the ketones as an energetic resource [24]. For instance, butyrate is typically produced in the intestinal epithelium and is usually present in the blood at very low concentrations. Butyrate metabolites in the blood can regulate physiological processes in the host organism [25]. It is known that butyrate exerts a local anti-inflammatory effect on the mucous membrane of the intestine and participates in stimulating regulatory T cells [26]. In another study, depletion of taxa in the *Lachnospiraceae* family was identified, as well as a low abundance of the *Eubacterium hallii* group, which are producers of butyrate. Significant associations were found between alpha-diversity or genes encoding butyrate-acetoacetate-CoA transferase and clinical or hemodynamic markers of HF. Higher levels of *Prevotella*, *Hungatella*, and *Succinclasticum* were observed [27]. Increased expression of acetate by the gut microbiota stimulates the activation of the nervous system and cognitive functions, increases appetite, and contributes to obesity. Converting acetate into other SCFAs, such as butyrate, may represent another way to regulate obesity and hyperinsulinemia [28,29]. Bacterial translocation and expression of bacterial products from the gut microbiome can impact the body. In particular, many metabolites are absorbed into the systemic circulation and can affect organs directly or indirectly [30,31].

### 2.2. SCFAs and Gut–Heart Axis

The gut–heart axis refers to the two-way communication system between the gastrointestinal tract and the cardiovascular system, with SCFAs playing an important role in this interaction.

The primary mechanisms of SCFA action in the gut–heart axis involve G-protein-coupled receptor (GPCR) activation, histone deacetylase inhibition, and mitochondrial function restoration [32,33,34]. These pathways collectively contribute to cardiac function regulation and cardiovascular health maintenance. HF-induced gut hypoperfusion and congestion lead to dysbiosis, reduced SCFA production, and increased intestinal permeability [35].

Among SCFAs, butyrate and propionate demonstrate particular significance through blood pressure reduction, improvement in ischemia-reperfusion injury, and decreased risk of atherosclerosis [10]. In heart failure, these SCFAs act as energy substrates while also inhibiting histone deacetylase-regulated gene expression and activating GPCR signaling, ultimately enhancing cardiac function [36]. The anti-inflammatory properties of SCFAs also contribute to inhibiting cardiac fibrosis, a key pathological process in heart failure development [37].

Studies have shown decreased microbial richness and diversity in HF patients compared to healthy controls [38]. Notable changes include reduced abundance of SCFA-producing bacteria, including *Lachnospiraceae*, *Faecalibacterium*, and *Eubacterium* species [32]. These alterations result from reduced intestinal perfusion and congestion in HF, leading to hypoxic conditions, altered nutrient absorption, and compromised barrier function [39,40] The disrupted gut environment can result in bacterial translocation and increased production of harmful metabolites like TMAO, further contributing to HF progression.

Therapeutic strategies targeting the gut–heart axis in HF include dietary modifications, pre-/probiotic administration, and potential fecal microbiota transplantation. Further research is necessary to establish the efficacy of these interventions [32]. Understanding the complex interactions between SCFAs and the gut–heart axis provides new opportunities for developing targeted therapies in heart failure management.

### 2.3. SCFAs and Related Metabolic Pathways in Heart Failure

In heart failure, SCFAs interact with various metabolic pathways through complex regulatory networks. These interactions include microbiota-derived metabolites such as TMAO, AAA derivatives, B vitamins, and BAs, all of which collectively affect cardiac function and disease progression. Understanding these metabolic connections is essential for developing targeted therapeutic approaches (Figure 1).

#### 2.3.1. SCFAs and TMAO Interaction in Heart Failure

SCFAs and TMAO exhibit opposing effects in heart failure pathophysiology. While SCFAs maintain intestinal barrier function and exert anti-inflammatory effects, TMAO, produced through microbial metabolism of dietary components (choline, phosphatidylcholine, l-carnitine, lecithin, betaine, and dimethylglycine), promotes disease progression [41].

The protective mechanisms of SCFAs include NLRP3 inflammasome inhibition and AMPK activation [42,43]. These actions protect against barrier disruption and metabolic stress while also promoting cholesterol excretion and gut homeostasis [44]. SCFA effects are mediated through histone deacetylase inhibition and GPCR activation [45].

The gut microbiota metabolizes TMAO through a complex network involving specific bacterial species. Enterobacteriaceae drive TMAO retroconversion to TMA while *Lachnoclostridium* and *Clostridium* species efficiently produce TMAO from choline [46]. Even low colonization levels of TMA-producing bacteria significantly affect TMAO accumulation [47].

In heart failure patients, elevated TMAO levels correlate with decreased SCFA-producing bacteria [48]. This imbalance triggers inflammatory cascades, increasing pro-inflammatory cytokines IL-6, TNF-α, and IL-1β [49,50]. TMAO levels positively correlate with inflammatory markers CRP and MCP-1, while SCFAs demonstrate opposing effects [51,52]. The study also found that TMAO levels were positively correlated with levels of inflammatory markers, such as C-reactive protein (CRP) and monocyte chemoattractant protein-1 (MCP-1).

Patients with heart failure with reduced ejection fraction (HFrEF) exhibit particularly high levels of both pro-inflammatory cytokines and TMAO [53,54]. Microbiome analysis reveals Actinobacteria enrichment and *Bifidobacterium* abundance, with positive correlations between *Escherichia*/*Shigella* genera and both TMAO and indoxyl sulfate levels [55]. These compositional changes directly influence the SCFA-TMAO balance, affecting heart failure progression.

#### 2.3.2. SCFAs and Aromatic Amino Acid Metabolism in Heart Failure

The interaction between SCFAs and aromatic amino acid metabolism represents a crucial regulatory network in heart failure, where bacterial species simultaneously participate in both metabolic pathways. *Actinobacteria*, *Firmicutes*, *Bacteroidetes*, and *Proteobacteria* orchestrate the coordinated production of SCFAs and metabolism of tryptophan, tyrosine, and phenylalanine [56,57,58,59,60].

SCFAs and tryptophan derivatives demonstrate complex metabolic interactions through shared regulatory pathways. Their synergistic effects on host inflammatory, immune, and metabolic responses involve multiple mechanisms [61,62]. SCFAs enhance tryptophan metabolism through modulation of bacterial enzymes, while tryptophan metabolites can influence SCFA production pathways. This metabolic coupling enhances intestinal homeostasis and barrier function, with tryptophan catabolites activating the aryl hydrocarbon receptor (AhR) to strengthen epithelial integrity and immune responses.

The bacterial species involved in these metabolic pathways exhibit complex cross-feeding relationships. Bacteria that produce SCFAs often supply essential nutrients for organisms that metabolize AAAs, while metabolites resulting from AAA processing can promote SCFA production. This metabolic cooperation is disrupted in heart failure, leading to imbalanced production of both beneficial and harmful compounds.

Indole-3-propionic acid (IPA) exemplifies this beneficial interaction, demonstrating significant cardioprotective effects through multiple mechanisms [63]. Through aryl hydrocarbon receptor binding, IPA regulates multiple pathways critical for cardiac function, including modulation of energy metabolism, reduction of oxidative stress, attenuation of cardiac fibrosis, and improvement of diastolic function. These effects complement SCFA-mediated cardioprotection.

Clinical studies have shown that disruption of both SCFA production and AAA metabolism significantly impacts heart failure outcomes. In heart failure patients, dysbiosis disrupts these metabolic pathways [64], leading to decreased levels of beneficial metabolites and accumulation of harmful compounds. SCFAs, particularly propionate, improve lipid metabolism and reduce hypertensive cardiovascular damage through specific membrane receptor activation and Nrf2 signaling pathway induction [65,66]. Complementarily, IPA protects against heart failure with preserved ejection fraction (HFpEF) by regulating the nicotinamide adenine dinucleotide salvage pathway and SIRT3 expression [67].

The disruption of these pathways leads to increased harmful metabolites, notably indoxyl sulfate (IS). Elevated plasma IS levels predict cardiovascular events in chronic heart failure patients [68], particularly affecting hemodialysis patients [69]. IS exerts pro-fibrotic, pro-hypertrophic, and pro-inflammatory effects through MAPK and NFκB pathway activation [70]. These effects are exacerbated by reduced SCFA production, creating a cycle of metabolic dysfunction.

The SCFA-phenylacetylglutamine (PAGln) axis represents another critical interaction in heart failure pathophysiology. While SCFAs demonstrate protective effects, elevated PAGln correlates with increased heart failure risk [71]. The microbial ecology underlying these interactions involves significant shifts in bacterial communities producing both SCFAs and aromatic amino acid metabolites. For example, the enrichment of certain bacteria (*Roseburia*, *Ruminococcus*, *Romboutsia*, and *Blautia*) affects both SCFA production and PAGln metabolism [45,46]. These compositional changes in the microbiota suggest potential therapeutic approaches targeting both metabolic pathways [72].

#### 2.3.3. SCFAs and B-Vitamin Synthesis in Heart Failure

SCFA-producing bacteria in the gut microbiome often simultaneously synthesize B vitamins through interconnected metabolic pathways. This metabolic coupling between SCFA and B-vitamin production represents a key mechanism by which gut bacteria contribute to both microbiota and host health [73]. Key bacterial phyla such as Firmicutes and Bacteroidetes demonstrate dual metabolic capabilities, producing both SCFAs and B vitamins through coordinated pathways.

The metabolic interconnection manifests through shared bacterial processes. For example, Lachnospiraceae family members, known for significant butyrate production, also contribute substantially to B-vitamin synthesis [74]. The production of SCFA and B vitamins is linked through common metabolic pathways. In particular, the metabolic pathways involved in butyrate production provide the necessary precursors for vitamin B synthesis. This sequential production promotes efficient energy use and optimal metabolic performance.

Within the Firmicutes phylum, notable SCFA producers also have significant capabilities for synthesizing B vitamins. Representatives of *Lachnospiraceae* and *Ruminococcaceae* produce complex vitamins such as folate (B9) and cobalamin (B12) while maintaining high SCFA production [73,75,76,77].

This metabolic cooperation becomes particularly crucial given that heart failure patients often show deficiencies in both compounds [78,79,80,81].

The therapeutic implications of this SCF–B-vitamin relationship extend beyond basic metabolic support. B-group vitamins (B1/B2/B6/B12), working synergistically with SCFAs, contribute to improved energy metabolism, reduced oxidative stress, decreased cardiomyocyte apoptosis, and reduced myocardial fibrosis.

These combined effects help optimize cardiac function by enhancing blood flow, improving heart contractility, and supporting overall cardiovascular health [82,83].

#### 2.3.4. SCFA and BA Interactions in Heart Failure

Recent research highlights the complex interactions between SCFA and BA metabolism in heart failure pathogenesis. SCFAs play a crucial role in regulating cardiac function, systemic immunity, and metabolism while also serving as an effective fuel source for the failing heart [36]. The gut microbiome influences the production of both SCFAs and BAs, creating an intricate metabolic network that affects cardiovascular health [84,85,86].

BAs are synthesized in the liver from cholesterol through biosynthesis, where cholesterol is converted into cholecystokinin, then into cholic acid, and subsequently transformed into primary BAs such as cholic acid or chenodeoxycholic acid. These primary BAs can then be converted into secondary BAs in the gut through microbial action, which occurs alongside SCFA production [87]. The bacteria responsible for SCFA production, particularly species of *Firmicutes* and *Bacteroidetes*, often participate in both metabolic pathways.

The interaction between SCFAs and BAs becomes particularly significant in heart failure through multiple mechanisms. While SCFAs can activate pathways that increase long-chain fatty acid oxidation and improve overall energy availability [37]. BAs act as signaling molecules through specific receptors such as the farnesoid X receptor (FXR) and Takeda G protein-coupled receptor 5 (TGR5) expressed in the heart [88]. This dual regulation appears important for maintaining cardiac function.

Studies in chronic heart failure patients have revealed important disruptions in these metabolic pathways. Mayerhofer et al. [89] demonstrated that patients with chronic heart failure had lower levels of primary BAs and higher ratios of secondary to primary BAs compared to healthy controls. This altered BA profile was associated with reduced overall survival, with approximately 40% of patients with the highest tertile ratio of secondary to primary BAs dying during the 5.6-year follow-up period. These changes often coincide with disturbed SCFA production, suggesting a linked metabolic disturbance [89,90].

The disruption of both SCFA and BA signaling pathways may contribute to cardiac dysfunction through multiple mechanisms, including impaired contractility and hypertrophy. Current research suggests that increased levels of secondary BAs and decreased SCFA production may contribute to systemic inflammation and oxidative stress observed in chronic heart failure [90]. Identification of interactions between SCFA producers and BAs metabolism opens new therapeutic possibilities. However, further studies are needed to elucidate the mechanisms of these interactions and their clinical implications [91,92].

## 3. Short-Chain Fatty Acids and Their Association with Biomarkers of HF

### 3.1. Relationship of SCFAs and HF

Certain types of gut bacteria are associated with the development and progression of CVD [93,94]. HF is the main cause of lethal outcomes in the case of most heart diseases. The recent scientific directions in the HF studies are focused on microbiome alterations, precisely bacteria producing SCFAs [9,10,95,96], which can be a prospective field in HF diagnosis, treatment, and prognosis. Thus, Sun et al. demonstrated the peculiar microbiome shifts in HF patients, precisely the depletion in number of SCFA-producing bacteria, such as *Ruminococcaceae* and *Lachnospiraceae* genera [97]. The gut microbiome can influence the physiology of the body through various mechanisms. The imbalance of the microbiota is the main cause of decreased levels of butyrate in the intestine and is associated with the development of multiple sclerosis and cardiovascular diseases. The study conducted by researchers reported significant differences in the bacterial composition of the gut microbiota in patients with severe HF. The Proteobacteria type was found to be significantly prevalent, while the *Firmicutes* type was significantly reduced. In the microbiota of HF patients, there was a decrease in the abundance of the *Ruminococcaceae*, *Lachnospiraceae*, and *Dialister* genera and an increase in the abundance of the *Enterococcus* and *Enterococcaceae* genera [97]. There are numerous factors associated with an increased risk of cardiovascular diseases; however, an increasing amount of data indicates the significant role of the gut microbiome and metabolites in the development of such diseases. Conversely, the beneficial effects of SCFAs, synthesized from dietary fibers by gut bacteria, are widely documented in the literature. SCFAs not only serve as a crucial energy substrate but also exhibit a range of protective effects by modulating anti-inflammatory cascades and improving metabolic homeostasis. These findings underscore the potential therapeutic value of SCFAs in the treatment of CHF [98,99,100].

However, there are a handful of studies describing the direct impact of SCFAs on the progression of cardiovascular diseases. The study conducted by Peng et al. found that the microbial communities in the human gut primarily belong to *Proteobacteria* and *Actinobacteria*. At the genus level, the HF group demonstrated higher levels of *Shigella*, *Bacteroides*, *Streptococcus*, *Gemmiger*, and *Akkermansia*, while lower levels were observed for *Lactobacillus*, *Bifidobacterium*, and *Enterococcus*. In addition, an assessment of SCFA concentrations in feces showed that levels of isobutyric acid and acetic acid were lower in patients with HF compared to those in the control group. According to correlation analysis, *Tenericutes* and *Bacteroidetes* showed positive correlations with most SCFAs. Furthermore, *Verrucomicrobia* positively correlated with valeric acid and isovaleric acid, while the abundance of *Chlamydiae* positively correlated with caproic acid [101].

Present studies assume that SCFAs have a protective effect against cardiovascular diseases, although the direct influence of SCFAs on HF is not fully understood. However, in animal studies it has been found that intake of fiber and acetate reduces the ratio of heart weight to body weight, decreases hypertrophy, and improves heart function [102]. Experimental data in mice have shown that administration of butyrate preserves cardiac function after myocardial infarction in the presence of intact gut microbiota. It is known that bacteria producing butyrate improve heart condition, providing new insights into the gut–heart axis during cardiac recovery [103]. The authors noted that in patients receiving probiotic intervention, there was an increase in the ability to produce SCFAs, which correlated with a decrease in systolic blood pressure (SBP), diastolic blood pressure (DBP), and high-sensitivity C-reactive protein (hsCRP) levels, reaching significance. These changes in blood pressure correlate with shifts in potential SCFA production, indicating a potential link between gut microbiota modulation, SCFA production, and blood pressure regulation [104]. Thus, SCFA production capacity may be a useful indicator for assessing the state of the host-microbiome interface in patients with cardiovascular diseases.

### 3.2. Left Ventricular Ejection Fraction and Its Association with SCFAs in HF

LVEF is a key parameter for assessing heart function in HF. This measure quantifies the percentage of blood ejected from the left ventricle during each cardiac cycle. Normal LVEF values typically range from 55 to 70%. In HF the LVEF may be reduced due to impaired contractility of the heart muscle or enlargement of the ventricle. This indicates a decline in cardiac function and can be used for classifying the severity of HF and monitoring treatment effectiveness. It is important to note that LVEF is just one of several parameters used to evaluate cardiac function in HF [86,105,106,107].

For many years, ejection fraction from the left ventricle has been widely used as a standard indicator of the function of this part of the heart in various cardiovascular diseases and medical procedures. Decreased heart function due to HF can result in reduced blood flow to the intestines, leading to intestinal tissue receiving less oxygen, which may cause tissue damage over time [108]. As a result of these changes in intestinal barrier function, increased intestinal permeability, development of intestinal insufficiency, bacterial translocation, and elevation of circulating endotoxins may occur, potentially triggering inflammation associated with cardiovascular diseases [39].

Studies have reported reduced LVEF as an indicator of HF. Patients with HF exhibited altered structure and function of the intestines, resulting in a significant decrease in gut microbiota diversity. Additionally, it was reported that *Romboutsia* and *Blautia*, *Eubacterium*, and *Ruminococcus* were positively correlated with LVEF [109]. Enrichment of the genus *Bradyrhizobium*, *Sphingomonas*, and *Sphingosinicella* was negatively correlated with LVEF levels. Concentrations of acetate, propionate, and butyrate in feces were significantly lower in individuals with HF. The decrease in SCFA concentrations in feces was associated with a reduction in major SCFA-producing bacteria. Furthermore, the concentration of acetate, propionate, and butyrate in stool samples positively correlated with LVEF [110,111].

SCFAs produced by the gut microbiota may help improve or preserve ejection fraction in HF through mechanisms such as enhancing cardiac energy utilization, reducing hypertrophy/fibrosis, and improving vascular function. SCFAs such as butyrate, acetate, and propionate may play a protective role in HF with HFrEF. It has been found that in patients with HFrEF, the abundance of bacteria producing butyrate, such as *Lachnospiraceae*, suggests that lower levels of SCFAs may contribute to the pathogenesis of HFrEF [112]. According to present studies, butyrate enhances mitochondrial adenosine triphosphate (ATP) synthesis and contractile function in a rat model of metabolic heart disease, potentially rectifying energy starvation in the damaged heart. In a mouse model, propionate has been shown to reduce myocardial hypertrophy, fibrosis, and vascular dysfunction, which may help preserve ejection fraction. SCFAs can enhance myocardial energy metabolism through mechanisms such as increasing the activity of the enzyme ACSM3, which promotes the oxidation of butyrate to produce ATP in the impaired heart. A high-fiber diet supplemented with acetate prevented the development of hypertension and HF in mice with hypertension, likely mediated by the beneficial effects of SCFAs [113,114].

In summary, SCFAs produced by the gut microbiota may help improve or preserve ejection fraction in HF through mechanisms such as enhancing cardiac energy utilization, reducing hypertrophy/fibrosis, and improving vascular function [33,115].

### 3.3. NT-proBNP and Its Association with SCFAs in HF

To our knowledge, the specific correlation between NT-proBNP and SCFAs in HF has not been thoroughly studied. NT-proBNP is a marker of cardiac stress; its levels are often elevated in HF, reflecting increased strain on the ventricular wall and volume overload [116]. On the other hand, SCFAs are primarily produced as a result of the fermentation of dietary fibers by the gut microbiota and exert various beneficial effects on cardiovascular health, including potential anti-inflammatory and metabolic effects. While increasing evidence supports the involvement of the gut microbiota and its metabolites, including SCFAs, in the pathophysiology of HF, their specific association with NT-proBNP levels in HF patients remains unclear and requires further investigation.

Recent research indicates that stable symptomatic HF patients with elevated NT-proBNP levels demonstrate alterations in gut microbiota composition. Specifically, there is a noted reduction in the prevalence of *Prevotella*, *Bifidobacterium*, *Parasutterella*, *Coprobacter*, *Ruminococcus gauvreauii*, and *Clostridium methylpentosum*, coupled with an increased relative abundance of *Clostridium sensu stricto* and *Veillonella*. In patients experiencing acute myocardial infarction, a positive correlation was observed between NT-pro-BNP levels and the genus *Clostridium innocuum*. Conversely, NT-pro-BNP levels showed a negative correlation with the genera *Weissella* and *Veillonella*. Functionally, several of these taxa are recognized as producers of SCFAs, including acetate and butyrate. However, no statistically significant associations were identified between NT-proBNP levels and alpha diversity, nor between LVEF and alpha diversity. Nonetheless, patients with NT-proBNP levels exceeding the median exhibited significantly elevated TMAO levels [117,118]. In HF patients, higher levels of the genera *Bradyrhizobium*, *Sphingomonas*, and *Sphingosinicella* were found to positively correlate with NT-proBNP levels and negatively correlate with LVEF. In terms of SCFAs, the presence of butyrate in stool samples was inversely associated with NT-proBNP levels in plasma. Additionally, there was a negative correlation between acetate levels and NT-proBNP, whereas acetate levels positively correlated with LVEF [110]. In the study by Yang et al., NT-proBNP demonstrated a negative correlation with *Blautia* and *Ruminococcus*, while it showed a positive correlation with *Phascolarctobacterium* [119].

It has also been found that elevated levels of NT-proBNP negatively correlate with LVEF, and their levels increased with improved cardiac function regardless of the cause of HF. The inverse relationship between CRP, NT-proBNP levels, and LVEF indicates that a higher level of NT-proBNP usually corresponds to a lower LVEF, indicating a more severe condition of HF [120]. In the study by Matin et al., the authors investigated the impact of probiotic consumption on levels of NT-proBNP, hsCRP in serum, and blood pressure in patients with CHF. The study results provide insights into the potential therapeutic effects of probiotics in the treatment of CHF, shedding light on their influence on important biomarkers such as NT-proBNP, as well as blood pressure levels [121]. Thus, it is conceivable that future research may explore potential correlations between SCFA levels and NT-proBNP in patients with HF to better understand the interaction between gut microbiota, SCFAs, and cardiac function in this condition.

### 3.4. Glomerular Filtration Rate and Its Association with SCFAs in HF

It is known that renal function impairment goes along with the CHF morbidity, or cardiovascular risk increases with the progression of CKD [122,123,124]; moreover, present studies underline the “gut–kidney axis” as one of the key points in the kidney’s function regulation through the metabolic and immune interactions [125]. It is known that changes occur in the composition of bacteria synthesizing SCFAs during the progression of CKD. This raises questions about the potential role of these metabolites in the progression of CKD and its associated cardiovascular complications [126,127]. These changes can influence inflammatory processes, immune response, and metabolic health, making SCFAs potentially significant in the context of treating and preventing complications related to CKD. The accumulated studies based on SCFAs effect on GFR demonstrate the negative trend between these two parameters [128,129]. Zhong et al. report the significantly diminished levels of propionate, acetate, and butyrate in patients with low kidney function [128]. Wang et al. reported the decrease of butyrate level in patients with impaired kidney function and the negative correlation between butyrate level and renal function [129]. It is known that various representatives of the gut microbiome are able to produce SCFAs, yet the major effect manifests due to interactions between SCFAs and multiple receptors within and out of the gut. GPR41 and GPR43 receptors are G-protein-coupled receptors (GPCRs) that are primarily involved in the recognition of SCFAs, which are produced by the gut microbiota during the fermentation of dietary fibers. These receptors are expressed in various tissues, including the gut, adipose tissue, immune cells, and endothelial cells, and play an important role in regulating various physiological processes, including inflammation, metabolism, and blood pressure regulation [130]. The connection between GPR41/GPR43 receptors and hypertension is increasingly being studied due to their role in regulating the sympathetic nervous system, inflammation, and vascular tone—all of which are crucial factors in the development of hypertension. Their activation may potentially provide a protective effect against hypertension, suggesting that enhancing SCFA production through diet or modulation of the microbiota could be an effective approach for managing or preventing high blood pressure [131]. SCFAs have been shown to increase the expression of the GPR43 protein in glomerular mesangial cells (GMCs), which are crucial for the structure and function of the kidney’s filtering units, the glomeruli. In conditions that mimic diabetic nephropathy, like high glucose levels and lipopolysaccharide (LPS) exposure, SCFAs mitigate harmful effects by boosting GPR43 expression. This indicates that SCFAs may protect against inflammation and oxidative stress caused by hyperglycemia and endotoxins [132,133].

Studies show that serum butyrate levels positively correlate with eGFR, a key indicator of kidney function. Additionally, positive correlations are observed between serum levels of total SCFAs, including acetate and propionate, and eGFR levels. This indicates that higher levels of these SCFAs in the blood may be associated with improved kidney function or less kidney damage in patients with diabetic nephropathy [134].

Butyrate connects to specific receptors (GPCR for the bacterial fermentation product) on dendritic cells and leads to T cell activation and further anti-inflammatory cytokine secretion. T lymphocyte regulation is provided by propionate and acetate; by binding to certain receptors, these acids contribute to T lymphocyte differentiation, resulting in gut physiological and immunological function activation [135,136]. This way, on behalf of SCFAs, the pathological processes such as oxidative stress, inflammatory damage, and ischemia are under control [137]. The uremic toxins originating from gut microbiome also contribute to GFR decline [138,139]. Recent findings assume that SCFA depletion is associated with the elevation of a specific toxin, indoxyl sulfate, and its toxic effect on kidney tubular cells [140]. Thus, a recent study by Steenbeke et al. observed the decline in SCFA level in patients on dialysis [141], promoting the perspective to diminish uremic toxins by regulating gut microbiome compounds, precisely SCFAs. It was found that the SCFA produced by the commensal intestinal bacteria, specifically a significantly higher concentration of valerate in the plasma, is closely associated with cardiovascular diseases in patients with CKD, regardless of age, diabetes, hypertension, and statin use. Including valerate in the model with urine protein creatinine ratio (UPCR), GFR, hypertension, and diabetes significantly improved its effectiveness in identifying these diseases [142]. The accumulated data promote SCFAs as a new approach for the prognosis and treatment of HF by regulating the composition of the gut microbiome, and thus, determining SCFA level is considered promising.

## 4. Conclusions

Research has shown that SCFAs, which are metabolic products of bacteria, play a key role in regulating human metabolism. Meanwhile, dysbiosis of the microbiota and the accumulation of specific microbiota metabolites exacerbate the progression of HF through various pathways. Besides considering the presence of SCFAs in the gut, it is also important to account for their concentration in the blood and other tissues, as SCFAs can affect various organs and systems of the body. Studies have demonstrated that the production and consumption of SCFAs can improve or maintain LVEF in HF, significantly influencing the relationship between the microbiota and HF outcomes. Many studies focus on understanding the complex interaction between gut microbiota, SCFAs, and the pathophysiology of HF, offering insights into potential strategies targeting the gut–heart axis. The relationship between gut microbiota metabolites, particularly SCFAs and HF, requires further studies to fully elucidate the underlying mechanisms of their action and explore their potential as therapeutic targets and prognostic risk factors for the prevention and treatment of HF, establishing clear causal links. Elevated levels of NT-proBNP in HF are associated with changes in the composition of the gut microbiota, characterized by a decrease or increase in certain SCFA-producing taxa and certain metabolites, respectively. Changes in the composition of SCFA-synthesizing bacteria also occur with the progression of CKD and its associated cardiovascular complications. The accumulation of certain metabolites and uremic toxins originating from the gut microbiome also contributes to the decline in GFR. Overall, the connection between gut microbiota metabolites, especially SCFAs, and HF is complex and multifaceted. Further research is needed to fully determine the underlying mechanisms of their action and explore their potential as therapeutic targets and prognostic risk factors for the prevention and treatment of HF, establishing clear causal links.

## Figures and Tables

**Figure 1 biomedicines-13-00343-f001:**
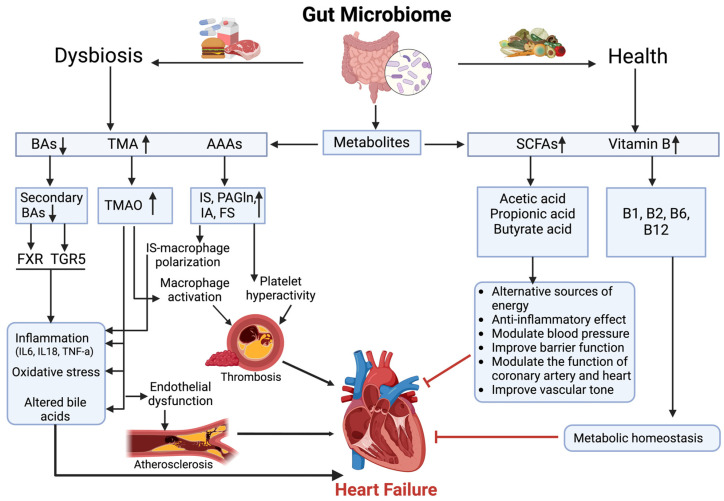
Role of microbiome metabolites in heart failure. An upward arrow (↑) indicates an increase in metabolite levels. A downward arrow (↓) indicates a decrease in metabolite levels.

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
