# Peer review of "Short-Chain Fatty Acids and Their Metabolic Interactions in Heart Failure"

_biomedicines, 2025, doi:10.3390/biomedicines13020343_

Round 1
Reviewer 1 Report
Comments and Suggestions for Authors
The manuscript requires more careful editing prior to submission.
The middle section, which focuses on "gut microbial metabolites and HF," appears to diverge from the stated focus of the title. To address this inconsistency, the authors should either narrow the scope of the manuscript to focus solely on SCFAs or revise the title to reflect the broader discussion on microbial metabolites in heart failure.
Additionally, the writing is overly verbose and repetitive. Please streamline the text to remove redundancies and improve the flow of ideas.
Comments on the Quality of English LanguageSome acronyms not defined
Author Response
Comments 1: [The manuscript requires more careful editing prior to submission.
The middle section, which focuses on "gut microbial metabolites and HF," appears to diverge from the stated focus of the title. To address this inconsistency, the authors should either narrow the scope of the manuscript to focus solely on SCFAs or revise the title to reflect the broader discussion on microbial metabolites in heart failure]
|
Response 1: Dear Reviewer, Thank you for your valuable comment regarding the discrepancy between the manuscript title and its content. We have carefully considered your suggestion and made the following changes: We have carefully revised the manuscript and made changes to the structure of Section 2 of the manuscript to strengthen the focus on SCFA in all sections. In addition, in response to your comment, subsection 2.3 has been substantially revised. Therefore, we propose revising the manuscript title to: "Short-Chain Fatty Acids and Their Metabolic Interactions in Heart Failure" This new title better reflects both: Primary focus on SCFAs Important metabolic interactions between SCFAs and other gut metabolites in the pathophysiology of heart failure An integrated approach to understanding these metabolic networks in the context of heart failure Additional changes: Strengthened the role of SCFAs in all sections where other metabolites are discussed Improved integration between SCFA-related pathways and other metabolic processes Updated references to include recent studies on SCFA interactions with other metabolites We believe these changes will improve the manuscript while maintaining the scientific merit and comprehensive nature of the review. The manuscript now presents a more focused look at SCFAs while contextualizing their important metabolic interactions in heart failure. Please let us know if any additional changes are needed. |
Comments 2: [Additionally, the writing is overly verbose and repetitive. Please streamline the text to remove redundancies and improve the flow of ideas] |
Response 2: Dear Reviewer, Thank you for your comment on the wordy and repetitive nature of the text. We have carefully revised the manuscript to improve its readability and flow. The following changes were made: a) The abstract was completely rewritten to be more concise and focused. b) Redundancy was eliminated throughout the manuscript by: removing duplicate descriptions of SCFA mechanisms across sections; consolidating information on metabolic pathways; and eliminating duplicate explanations of key concepts. These changes resulted in a more concise and focused manuscript while maintaining its scientific rigor. Please let us know if further optimization is required.
Comments 3: [Some acronyms not defined] Response 3. Dear Reviewer, Thank you very much for your accurate observation. In line with this, throughout the text, definitions have been provided for certain abbreviations, and conversely, abbreviations have been introduced for some definitions |

Reviewer 2 Report
Comments and Suggestions for Authors
Dear Sirs,
this manuscript deals with a very interesting issue, such as HF and SCFAs, as these are modified by the gut microbiome. The manuscript is very well written and presented. The references are more than adequate and up to date. My only suggestion in the following:
As hypertension is a major risk factor of HF, it would be helpful to the readers to add some more information about the GPR41/GPR43 receptors and their association with SCFAs in hypertension. This could be added in the section with "gut-kidney axis" referral by the authors. As this is matter of much interest and ongoing research, more information in this regard would be helpful and informative for the readers.
Author Response
Comments 1: [This manuscript deals with a very interesting issue, such as HF and SCFAs, as these are modified by the gut microbiome. The manuscript is very well written and presented. The references are more than adequate and up to date. My only suggestion in the following: As hypertension is a major risk factor of HF, it would be helpful to the readers to add some more information about the GPR41/GPR43 receptors and their association with SCFAs in hypertension. This could be added in the section with "gut-kidney axis" referral by the authors. As this is matter of much interest and ongoing research, more information in this regard would be helpful and informative for the readers]
|
Response 1: Dear Reviewer, Thank you very much for your positive feedback and for highlighting the importance of the topic addressed in our manuscript. We sincerely appreciate your valuable suggestion regarding the inclusion of additional information on the GPR41/GPR43 receptors and their association with SCFAs in hypertension, particularly in the context of the gut-kidney axis. In response, we have incorporated a detailed discussion of the role of GPR41/GPR43 receptors in hypertension in the lines 736-747, emphasizing their interaction with SCFAs and relevance to the gut-kidney axis. This addition can be found in the revised manuscript in the section titled "Gut-Kidney Axis." We believe this enhancement aligns well with the manuscript's focus and will provide additional insight for readers interested in this emerging area of research. Thank you once again for your constructive input, which has helped improve the manuscript.
|

Round 2
Reviewer 1 Report
Comments and Suggestions for Authors
A better resolutions for figure 1 is a must.
Author Response
Comments 1: [A better resolutions for figure 1 is a must]
|
Response 1: Dear Reviewer, Thank you for your valuable comments. We have carefully reviewed Figure 1 and agree that improving its resolution enhances the clarity and quality of the manuscript. To address this concern, we have:
Recreated Figure 1 with higher resolution using professional software Biorender. Adjusted the figure dimensions and resolution to meet the journal's specifications and improve visual quality. Verified that all text, labels, and graphical elements are sharp and legible. The updated figure has been included in the revised manuscript (Page 4, Figure 1). We hope this improvement meets your expectations, and we are happy to make further adjustments if needed. |